# Monitoring of Urban Black-Odor Water Using UAV Multispectral Data Based on Extreme Gradient Boosting

**Fangyi Wang [1,2], Haiying Hu [1], Yunru Luo [1,2], Xiangdong Lei [1,2,*], Di Wu [1,2] and Jie Jiang [1,2]**

1 State Key Lab of Subtropical Building Science, School of Civil Engineering and Transportation, South China University of Technology, Guangzhou 510641, China
2 Pazhou Lab, Guangzhou 510335, China
* Correspondence: ctleixiangdong@mail.scut.edu.cn

**Abstract:** During accelerated urbanization, the lack of attention to environmental protection and governance led to the formation of black-odor water. The existence of urban black-odor water not only affects the cityscape, but also threatens human health and damages urban ecosystems. The black-odor water bodies are small and hidden, so they require large-scale and high-resolution monitoring which offers a temporal and spatial variation of water quality frequently, and the unmanned aerial vehicle (UAV) with a multispectral instrument is up to the monitoring task. In this paper, the Nemerow comprehensive pollution index (NCPI) was introduced to assess the pollution degree of black-odor water in order to avoid inaccurate identification based on a single water parameter. Based on the UAV-borne multispectral data and NCPI of sampling points, regression models for inverting the parameter indicative of water quality were established using three artificial intelligence algorithms, namely extreme gradient boosting (XGBoost), random forest (RF), and support vector regression (SVR). The result shows that NCPI is qualified to evaluate the pollution level of black-odor water. The XGBoost regression (XGBR) model has the highest fitting accuracy on the training dataset ($R^2 = 0.99$) and test dataset ($R^2 = 0.94$), and it achieved the best retrieval effect on image inversion in the shortest time, which made it the best-fit model compared with the RF regression (RFR) model and the SVR model. According to inversion results based on the XGBR model, there was only a small size of mild black-odor water in the study area, which showed the achievement of water pollution treatment in Guangzhou. The research provides a theoretical framework and technical feasibility for the application of the combination of algorithms and UAV-borne multispectral images in the field of water quality inversion.

**Keywords:** black-odor water; unmanned aerial vehicle; extreme gradient boosting; machine learning





## 1. Introduction

Urban rivers are significant for urban development on account of their vital functions, including drainage, flood control, maintaining the regional water balance, and shaping the urban landscape. Recently, the acceleration of urbanization and the unbridled urban population have resulted in non-negligible water pollution problems in many cities [1,2]. China has successfully developed economic policies to enable a huge number of people to get escape poverty during the last decade, at the cost of severe environmental problems, including water quality degradation [3]. There are a number of contaminated rivers in the cities of China, some of which have formed black-odor water bodies [4,5]. The main reason why the black-stinking phenomenon appears is that the rivers have been heavily polluted by organic matter, heavy metals, or nutrients [6,7]. According to the Regulation Guide of Urban Black and Odorous Water Bodies (hereinafter referred to as "the Guide") released by the National Ministry of Housing and Urban-Rural Development, black-odor water is defined as water bodies with unpleasant colors and/or emitting stench. The black-odor rivers feature a decreasing number of aquatic organisms, and serious deterioration

of the structure and function of the river ecological system, which not only affects the image of the city, but also causes a serious threat to the health of urban residents and ecological security [8]. Therefore, the treatment of black-odor rivers has gradually become a prominent focus of environmental governance. In April 2015, Water Pollution Prevention Action Plan was officially launched, which explicitly stated that urban black-odor water bodies will be gradually eliminated by 2030. As of 31 December 2019, the total number of the existence of urban black and odorous water bodies identified in China was 2899 and the elimination rate in the whole country is 86.7% (https://www.mee.gov.cn/xxgk2018/xxgk/xxgk15/202001/t20200117_760049.html, accessed on 17 January 2020).

In this case, the advance in relevant technologies concerning the identification and monitoring of black and odorous water bodies is increasingly important. The conventional method for water pollution assessment is in situ monitoring, which can provide a relatively accurate evaluation of water quality with high investment and low frequency [9]. Pointwise surveys often fail to capture short-term temporal and spatial variations in water quality parameter. Conversely, the remote-sensing method is extensively used for its capacity to detect temporal variation and acquire spatial data on large scale [10,11]. The reflectance spectrum of water is determined by the optical properties of its constituents [12,13]. Hence, water with different kinds and levels of pollutants differs from spectral information and the pollution level can be characterized by water quality parameters, which can be applied to retrieve parameters indicative of water quality. Using satellite-based technology to measure water parameters has been conducted for many decades [14–18]. However, there are two main limitations of the application of satellite imagery. Such limitations include, on the one hand, the existence of atmospheric effects, which can keep the users from the information for extended periods, and on the other hand, its lack of adequate spatial resolution, especially for inland water bodies, such as urban rivers and reservoirs [19,20]. With the fast pace of technological development and the miniaturization of sensors, the unmanned aerial vehicle (UAV) remote sensing platforms were generated, i.e., systems integrated with UAV that can easily cover an area with complex terrain, portable spectral camera, matching software based in computer vision, and photogrammetry, providing products with ultra-high temporal and spatial resolutions [21,22]. Since UAV platforms can collect required high-resolution aerial images which are hardly affected by atmospheric effects [23], they offer a possible cost-effective and qualified solution to retrieve water quality parameters. In this case, UAV-borne remote sensing technology has been employed for the inversion of water quality parameters in recent studies [24–28]. Since optically active constituents in the water body alter the water leaving radiation by absorption and scattering characteristics for each constituent, the spectral properties of the water body allow one to obtain quantitative information on water constituents. There are three approaches to estimating the concentration of water quality indicators through remote sensing data: empirical, semi-empirical, and analytical approaches [29]. The analytical methods rely on the radiative transfer in the water column [30,31], which provides models with a physical basis. These models enable one to derive the optically active constituents such as Chl-a [32–35] or other indicators of water quality [36] using spectral images. However, the analytical methods require a great quantity of spectral data concerning optical constituents and complex analysis [37]. It is relatively easy to establish empirical models. Empirical approaches are conducted by establishing relationships between in-situ measurements of variables of interest and accompanying remote-sensing data, without physical explanations of the relationship and inherent and apparent optical characteristics of constituents. Then, the regression model is applied to every image element for the spatial distribution of the variables. The most widely used regression approach is linear regression. In the development of linear regression, the band ratio method and logarithmic transformation were adopted to process data (reflectance extracted from remote-sensing images and in-situ data), to strengthen the linear correlation among variables, which helps to propose best-fit models [38–41]. However, sometimes, the relationship between spectral information and measured values is nonlinear, in which case, the use of a nonlinear regression model can enhance the fitting accuracy

of the relationship [28,42,43]. Methods of nonlinear regression include statistical analysis regression and artificial intelligence algorithms. Due to the remarkable ability to explore potential connections between data and advantages, such as noise filtering [44], artificial intelligence algorithms achieve higher accuracy than statistical analysis regression most of the time. Random forest (RF) is one of the popular algorithms [45,46] because it can construct a robust model in the case of random disturbances and the problem of overfitting rarely occurs [47,48]. In the study of chromophoric dissolved organic matter inversion, the performance of a random forest regression (RFR) model with the lowest prediction error was superior to that of other regression models like the backpropagation neural network model [49]. Support vector regression (SVR) is robust for the noise data and can produce accurate results even with a small size of data, which makes it widely used in the field of water parameters and hydrological inversion [50–52]. Since the prediction accuracy of SVR is closely related to penalty factor, type of kernel function, and its adjustable parameters, research on it mainly focuses on the method of parameter optimization. Compared with traditional algorithms, such as RF and SVR, the emerging algorithms, including categorical boosting (Catboost), gradient boosting decision tree (GBDT), extreme gradient boosting (XGBoost), have the advantages of stronger generalization ability and simpler parameter adjustment. In the experiment where various machine learning algorithms were systematically evaluated for the retrieval of water quality parameters with UAV-borne hyperspectral data, the results demonstrated that the overall prediction accuracy of Catboost regression model was higher than that of other traditional machine learning models like RFR model [53]. For urban rivers, Wei et al. [54] applied the GBDT algorithm and other algorithms to establish inversion models based on UAV-borne hyperspectral information and in-situ measurements. The comparison of the models' performance indicated that the GBDT regression model achieved the highest retrieval accuracy efficiently. These studies show that emerging boosting algorithms are applicable in the field of quantitative remote sensing of water based on UAV-borne hyperspectral imagery. However, there is still a lack of research on retrieving more water quality parameters using UAV data and emerging artificial intelligence algorithms. XGBoost selected in our study employs gradient boosting to optimize loss function as Catboost and GBDT do. XGBoost is improved from GBDT by optimization, in which case, XGBoost can achieve better predictive accuracy. The potential of XGBoost algorithm for inverting a comprehensive index of water quality needs to be further verified.

Considering the method of determining black-odor water in the Guide [55], a comprehensive index is more appropriate to identify the existence of black-odor water and evaluate the pollution level instead of a single water parameter. Thus, the Nemerow comprehensive pollution index (NCPI) is selected. NCPI is a multi-factor weighted index that is originally applied for soil heavy metal pollution evaluation and the comprehensive assessment of environmental quality [56]. In this study, we applied a UAV integrated with a multispectral camera to obtain ultra-high resolution multispectral imagery. Simultaneously, the in-situ campaign was carried out to acquire the measured values of four water parameters, including ammonia nitrogen (AN), ORP (oxidation-reduction potential), DO (dissolved oxygen), and SD (depth of Sechi disk), based on which the NCPI of every sampling point can be calculated. Then, the band combinations and NCPI values for every sampling point are fed into the XGBoost algorithm, RF algorithm, and SVR algorithm to train the inversion models.

The objectives of this paper mainly are: (a) to establish models to retrieve NCPI based on UAV-borne multispectral data and different algorithms; (b) to verify the potential of the application of the XGBoost in the field of water parameter inversion; (c) to analyze the NCPI distribution in the study area for supporting polluted water monitoring and targeted treatment technically.

## 2. Methodology

### 2.1. Framework for NCPI Inversion Model

As shown in Figure 1, there are 4 steps in the framework for NCPI inversion model development. Firstly, the acquisition of UAV-borne data and in situ data were conducted simultaneously, based on which band combinations and NCPI were calculated. Then, 3 estimation models were established using band combinations and NCPI as input variables on the basis of the different algorithms. The best-fit model was selected by the means of the inversion accuracy assessment. The satisfying models were applied to map the spatial distribution map of NCPI based on the UAV-borne images.

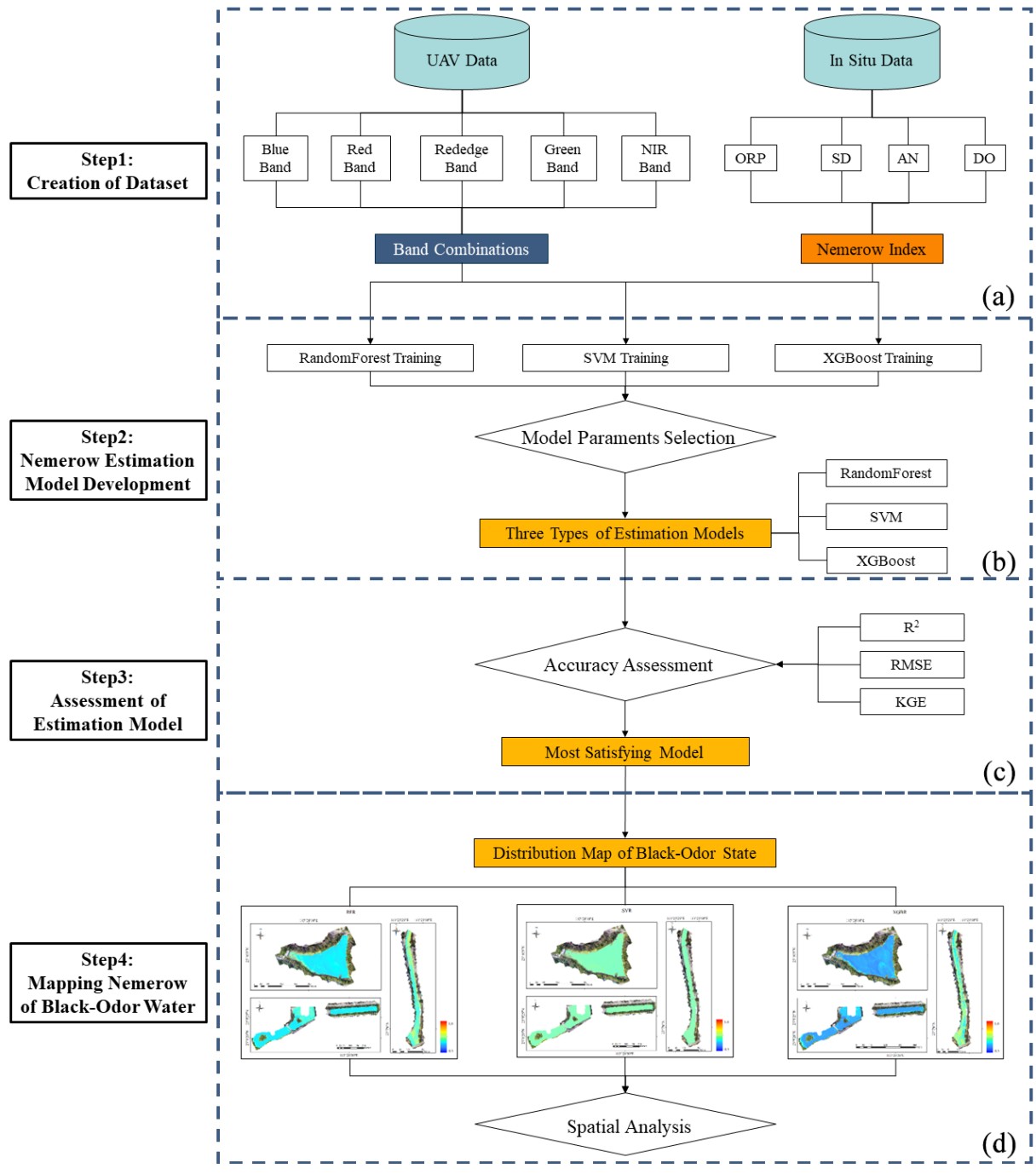

**Figure 1.** Framework of the NCPI inversion model developed in the study. (**a**) Step 1. (**b**) Step 2. (**c**) Step 3. (**d**) Step 4.

## 2.2. Study Area

This study area includes 5 regions all located in Tianhe District, Guangzhou, China. The climate of Guangzhou is marine subtropical monsoon, which is characterized by abundant precipitation and the precipitation in the summer half year is 70% of the annual precipitation. Guangzhou is situated in the lower reaches of the West River catchment, where the majority of rocks is sedimentary clasts and granites. Due to the relatively frequent geologic activities, thousands of years of agricultural development, high-density population, and long-term humid and hot environment, the chemical erosion process of silicate rocks in this basin is so strong that the chemical composition of water bodies is mainly impacted by rock weathering and human activities. The study region contains 4 lakes situated in the Wushan Campus of the South China University of Technology and they are named East Lake (23.16° N, 113.35° E), West Lake (23.16° N, 113.35° E), North Lake (23.17° N, 113.35° E) and Middle Lake (23.16° N, 113.35° E) respectively. The lakes have been under treatment for years and the pollution level has been reduced considerably. Hence, 41 evenly distributed sampling points were arranged for this area, as shown in Figure 2. The fifth region is a section of the low reaches of Chebei river (23.12° N, 113.40° E), which is a 656-m stretch from Yongtai Street to Jindong Road. The Chebei river is the longest river in the Tianhe District and has been included in the scope of black-odor water remediation. After effective governance, water in the Chebei river now can meet the class-III water standard for surface water. There are 11 evenly distributed sampling points along the stretch, as shown in Figure 2.

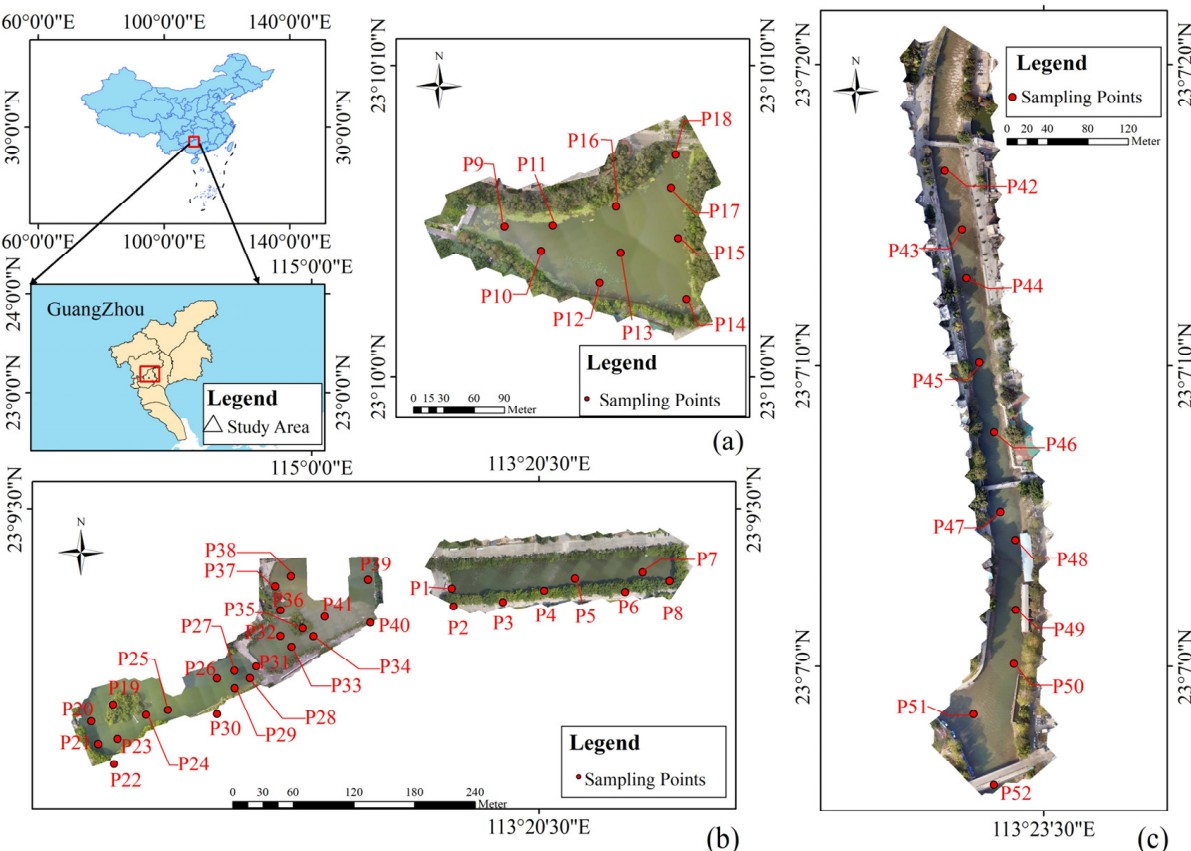

**Figure 2.** Sampling points in study area. (**a**) North Lake. (**b**) West Lake and Middle Lake (left); East Lake (right). (**c**) Chebei River.

The Tianhe District is the most economically developed area in Guangzhou with the most concentrated population and commerce. In this case, water quality in the Tianhe District has a significant impact on the development of Guangzhou. Since these 4 lakes represent one of the supply sources for the Liede river which flows through the Tianhe District, their water quality not only has a significant impact on the landscape of the area, but also affects the living condition of the residents along the river. The Chebei river carries out the duties of draining rainwater in the Tianhe District. The serious pollution problem of the Chebei river may result in poor water-carrying capacity. Due to their important geographical location and function, the local government attaches great importance to these 5 regions. Hence, selecting these 5 regions as the study area can not only support the government's planning, but can also allow to monitor the remarkable improvement of the water quality via remote-sensing after the previous remediation.

### 2.3. In Situ Data Collection

The data acquisition operation in 4 lakes lasted from 12 to 13 September 2021 and the survey conducted in Chebei river was on 7 January 2022. The in-situ data acquisition includes the collection of water samples and the measurement of ORP, DO, and SD. Water samples in bottles were brought back to the laboratory for the results of AN.

The method for the 4 parameters is completely according to the Guide. The acquisition of SD depends on the Secchi disk. When the disk is slowly sunk into the water until the black and the white color is indistinguishable, the scale value is recorded as the SD of this sampling point. However, it was the dry season when we surveyed the Chebei river. The measurement of water depth was adopted instead. DO was measured using a Lohand Biological LH-D701 portable dissolved oxygen meter. A Lohand Biological LH-M300 portable oxidation-reduction potentiometer was applied to measure ORP. Measurement of AN was conducted indoors based on a gas-phase molecular absorption spectrometer (Lohand Biological LHC660).

### 2.4. Airborn Multispectral Imagery Preprocessing

The instrument for remote-sensing usage is DJI Phantom 4 Multispectral (P4M) integrated multispectral imaging system. The imaging system integrates 1 red-green-blue (RGB) camera and a multispectral camera array with 5 cameras covering 5 bands, namely band 1 (450 nm ± 16 nm), band 2 (560 nm ± 16 nm), band 3 (650 nm ± 16 nm), band 4 (730 nm ± 16 nm), and band 5 (840 nm ± 26 nm). The cameras mounted on the UAV all have a resolution of 2 MP with a global shutter, on a 3-axis stabilized gimbal, which prevents photos from shaking due to un-expected factors like wind. The UAV features a Time Sync system and real-time kinematic (RTK) module, which work together to acquire real-time positioning data at the millisecond level.

Taking the condition of the study area into account, the flight plan was determined using the DJI GS Pro, which can plan a route automatically on the basis of the setting such as flight altitude, the front overlap ratio and side overlap ratio. With a flight height of 70–80 m in all 5 areas, the ground sampling distance is between 3.70 and 4.23 cm/pixel.

Considering the remote-sensing data initially collected are presented as digital number (DN) value per pixel in the image, essential preprocessing must be carried out to generate radiometrically calibrated reflectance maps. Routine operation of remote-sensing images preprocessing consists of radiometric correction, geometric correction, and mosaicking. In our study, preprocessing steps only include radiometric calibration and mosaicking, which are accomplished by DJI Terra as long as conversion parameters are offered. The method is described as follows:

(1) Radiometric correction consists of radiometric calibration and atmospheric correction. Since the flight height is so low that the atmospheric effect on the images can be neglected [53,57], the radiance from the sensor radiation calibration almost equals the surface reflectance. In addition, the models widely used for the atmospheric correction, such as the 6S atmospheric correction model and MODTRAN model, are not appropriate for the low-altitude situation. Consequently, atmospheric correction can be excluded from the radiation correction.

(2) The radiometric calibration, known as sensor radiation calibration, is meant to convert the digital measurement obtained by the sensor to actual radiance. This step demands three calibrated reference panels at different reflectance levels. Before each flight plan, these panels should be placed on the floor of the flight area and then UAV over the panels takes single-band photos of them. The images of panels in 5 regions of the electromagnetic spectrum and reflectivity data of panels are input to the DJI Terra to conduct radiometric calibration.

(3) Geometric correction attempts to correct positional errors and to transform the original image into a new image that has the geometric characteristics of a map since the geometric position and other characteristics of the ground features on the original image are often inconsistent with those of the corresponding ground features. In our study, RTK module integrated with P4M can provide every photo with centimeter-level and real-time positioning data. Due to the positioning error less than 0.01 m and the small size of study area, the geometric correction is ignored in the UAV image processing.

(4) Mosaicking is a method of constructing multiple images of the same scene into a larger image, whose output is the union of input images.

Considering the high resolution of the images from UAV, the mean window for reading the reflectance of the sampling point should not be a single pixel or the deviation will probably occur. Conversely, some features will be lost if the mean window is too large. Thus, a reflectance average of a $5 \times 5$ pixel matrix was chosen for the following experiment.

*2.5. Spectral Data Preprocessing*

In Section 2.3, the process that converts the UAV-borne images to reflectance maps was represented in detail. Spectrum information of 52 sampling points was extracted from the images of the study area. These 52 samples were randomly divided into a training set and a test set according to the ratio of 7:3, in which case, there were 37 training samples and15 test samples. The training data set was used to train the model based on the algorithm, and then the test data set served as a validation basis to derive performance criteria. The comparison of the inversion accuracy on the training data set and the test data set can determine whether the phenomenon of overfitting happens or not. By randomly dividing the training set and test set many times and modeling, the model with high fitting accuracy and strong generalization ability is selected. In the subsequent experiment, the band and band ratio highly correlated with the Nemerow Index needs to be found. Since the band ratio method can not only eliminate the interference of water surface roughness and background noise, but also enhance contrast in quantitative remote sensing inversion [58], the exhaustive method was employed to calculate the band ratio. Band combinations commonly used in satellite remote sensing were also adopted. Pearson correlation coefficients were determined to evaluate the correlation between band combinations and NCPI. The larger the coefficient, the stronger the correlation. The Pearson correlation coefficient was improved to 0.56 after the band ratio method, which was 0.41 before. Eventually, 33 features, including 5 single bands and 28 band ratios, were selected as variables to establish the inversion model. These 28 band combinations are shown in Table 1.

**Table 1.** Band combinations used for establishing inversion models.

| Band Combination (BC) | Band Math | Reference | Band Combination (BC) | Band Math | Reference |
|---|---|---|---|---|---|
| BC1 | B1/B2 | Simple ratio | BC15 | B4/B3 | Simple ratio |
| BC2 | B1/B3 | Simple ratio | BC16 | B4/B5 | Simple ratio |
| BC3 | B1/B4 | Simple ratio | BC17 | B5/B1 | Simple ratio |
| BC4 | B1/B5 | Simple ratio | BC18 | B5/B2 | Simple ratio |
| BC5 | B2/B1 | Simple ratio | BC19 | B5/B3 | Simple ratio |
| BC6 | B2/B3 | Simple ratio | BC20 | B5/B4 | Simple ratio |
| BC7 | B2/B4 | Simple ratio | BCB1 | $(B2 - B1)/(B2 + B1)$ | Normalized indices |
| BC8 | B2/B5 | Simple ratio | 3BDA | $(B3^{-1} - B4^{-1}) \times B5$ | [59] |
| BC9 | B3/B1 | Simple ratio | 3BDA_MOD | $(B3^{-1} - B4^{-1})$ | [60] |
| BC10 | B3/B2 | Simple ratio | NDCI | $(B4 - B3)/(B4 + B3)$ | [61] |
| BC11 | B3/B4 | Simple ratio | NDVI | $(B5 - B3)/(B5 + B3)$ | [61] |
| BC12 | B3/B5 | Simple ratio | SABI | $(B5 - B3)/(B1 + B2)$ | [62] |
| BC13 | B4/B1 | Simple ratio | KIVU | $(B1 - B3)/B2$ | [63] |
| BC14 | B4/B2 | Simple ratio | Kab1 | $1.67 - 3.94 \times \ln(B1) + 3.78 \times \ln(B2)$ | [64] |

*2.6. Modeling Approaches*

2.6.1. Nemerow Comprehensive Pollution Index

NCPI is a weighted multi-factor environmental quality index considering the prominent maximum value, in which case, the effect of the water quality parameter with extreme value on the water quality assessment can be emphasized [65]. According to the guide (Table 2), as long as a single indicator at a detection point reaches the severe level in Table 2, the point is regarded as severe black-odor water (SBO). This classification applies to mild black-odor water (MBO) and non-black-odor water (NBO). It can be found that the influence of a single indicator is highlighted in the evaluation of black-odor water, which is consistent with the characteristics of NCPI. Thus, NCPI is determined as the indicator of water quality in this experiment.

**Table 2.** Classification standard of the pollution levels of urban black-odor water.

| Characteristic Index | Mild | Severe |
|---|---|---|
| SD(cm) | 25—10 | <10 |
| DO(mg/L) | 0.2—2.0 | <0.2 |
| ORP(mV) | −200—50 | <−200 |
| AN(mg/L) | 8.0—15 | >15 |

The NCPI is defined as $0 < P_i \leq 1$ equals NBO; $1 < P_i \leq 2$ equals MBO; $2 < P_i \leq 10$ equals SBO, where $P_i$ stands for the NCPI of the i-th sample. The dimensionless linear relationship is shown in Figure 3. The horizontal axis node of the piecewise function is the classification threshold from Table 2.

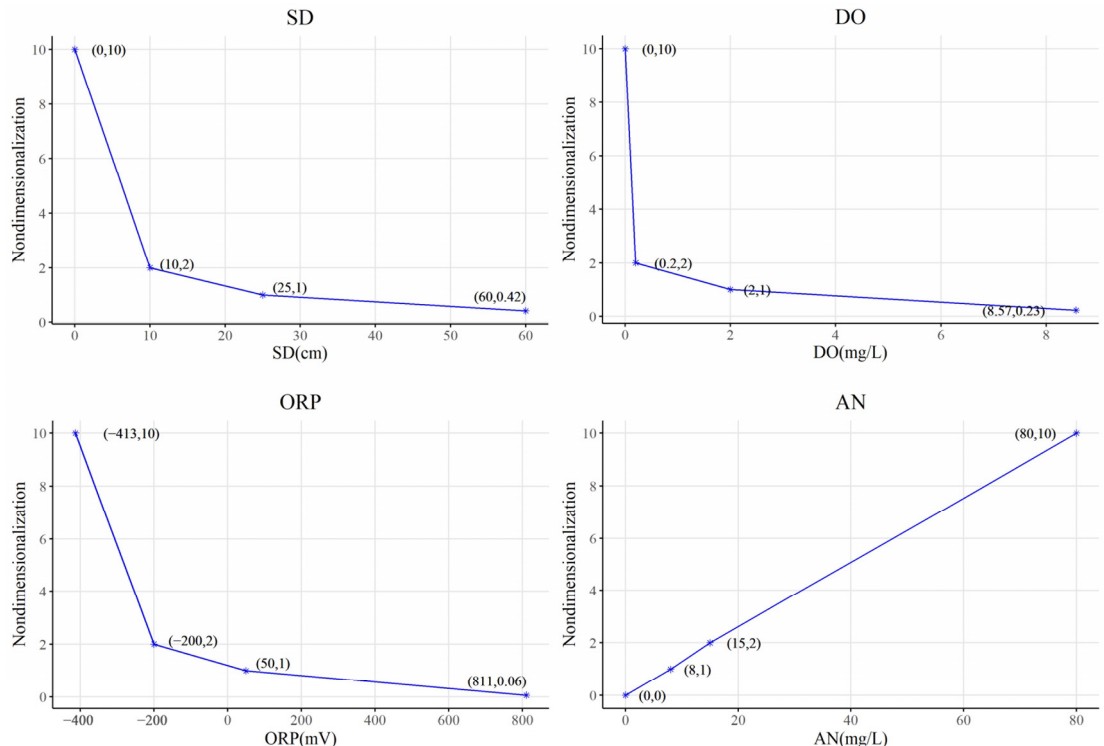

**Figure 3.** Dimensionless functions of SD, DO, ORP, AN.

The equation for the NCPI is as follows:

$$P_{final} = \sqrt{\frac{P_{max}^2 + \left(\sum\limits_{i=1}^{n} W_i P_i\right)^2}{2}} \qquad (1)$$

$$W_i = \frac{\frac{C_i}{S_i}}{\sum\limits_{i=1}^{n} \frac{C_i}{S_i}} = \frac{I_i}{\sum\limits_{i=1}^{n} I_i} \qquad (2)$$

where $P_{max}$ represents the maximum of all the indices $P_i$; $W_i$ indicates the weight of the water parameter in the i-th detection point; $I_i$ is the ratio of the i-th water quality parameter factor $C_i$ to its objective concentration $S_i$. The objective concentrations are from different criteria of water quality [54]. $S_{DO}$ (5 mg/L) and $S_{AN}$ (1 mg/L) are both obtained from the class-III water standard for surface water; $S_{SD}$ is set to 1.2 m according to the class A or B landscape-water standard. $S_{ORP}$ is decided as 50 mV from the classification standard of the pollution degree of black-odor water.

### 2.6.2. Extreme Gradient Boosting Regression and Other Models

The ensemble learning is to create a series of base learners most of which are weak learners, and then combine them to form a more comprehensive strong learner. As one of the ensemble algorithms, boosting algorithms generate base learners sequentially, in which case, the weight of samples is updated according to the learning error rate of previous base learners (increasing the weight of the wrongly classified samples) every time new learners are trained. The process is repeated until the final strong learner is combined with a certain number of base learners. As one of boosting algorithms, GBDT algorithm features constructing a decision tree as a base learner in each iteration based on the negative gradient direction of the model loss function [66,67]. XGBoost algorithm was originally proposed by Tianqi Chen based on the optimization of GBDT [68]. There are

two main improvements. Firstly, compared to GBDT, a regularization term is added to the objective function expression of XGBoost to control the complexity of the model, which is beneficial to preventing overfitting and improving the generalization ability of the model. Secondly, XGBoost employs the second-order Taylor expansion of the objective function to quickly optimize the objective in general while GBDT only uses the first-order derivative information of the cost function during model training. The optimization of computational efficiency and the model's generalization ability supported XGBoost to outperform GBDT.

Hyperparameter eta represents shrinkage coefficient of each tree, in which case, it can reduce the influence of each individual tree and leave space for future trees to improve model. The range of eta is (0, 1). The larger the value of eta, the fewer iterations and the easier it is too overfit, in which case, the global optimal solution may be missed and it is difficult to globally converge the algorithm. The learning rate eta should have a lower bound or the number of iterations would be too large when achieved the same accuracy, which leads to a waste of computational time. Hyperparameter gamma stands for the minimum loss reduction required for leaf node splitting. The larger this value is, the more difficult the leaf node splitting is and the more conservative the model is. Thus, gamma can control the model complexity so that it plays a vital role in preventing overfitting. The range of gamma is [0, +∞]. The smaller the value of gamma, the higher the model complexity and the easier it is too overfit. In order to prevent overfitting, we tried to adjust eta, gamma, and the number of iterations. Since the amount of sample is small, there is no additional adjustment for other parameters like the fraction of samples used to train the individual base learners (subsample), the maximum depth of the tree, and so on.

In the experiment, besides the XGBoost regression (XGBR) model, the RFR model and the SVR model are also trained to derive the statistical relationship between and target NCPI and band combinations. RF algorithm is a type of ensemble algorithms and it chooses a decision tree as its base learner. On the basis of building bagging ensemble learning, RF further introduces random attribute selection in the training process of a decision tree. SVR is a branch of support vector machine (SVM) and it can be obtained by extending SVM from classification problem to regression problem. The difference between SVR and SVM is that there is only one kind of sample point in SVR. The optimal hyperplane it seeks is not the "most open" of two or more kinds of sample points, but the minimum total deviation of all sample points from the hyperplane.

### 2.6.3. Model Evaluation

In this study, the coefficient of determination ($R^2$), root mean square error (RMSE), and mean absolute error (MAE) were selected to evaluate the performance of models. These indicators are defined as follows:

$$R^2 = 1 - \frac{\sum\limits_{i=1}^{N} (y\_\text{actual} - y\_\text{predicted})^2}{\sum\limits_{i=1}^{N} (y\_\text{actual} - y\_\text{mean})^2} \tag{3}$$

$$RMSE = \sqrt{\frac{\sum\limits_{i=1}^{N} (y\_\text{actual} - y\_\text{predicted})^2}{N}} \tag{4}$$

$$MAE = \frac{1}{N} \sum\limits_{i=1}^{N} |y\_actual - y\_predicted| \tag{5}$$

where *y_actual*, *y_predicted*, *y_mean* is real value, predicted value and the real mean value. *N* denotes the number of samples used to calculate accuracy. The value of $R^2$ ranges from 0 to 1. The closer its value is to 1, the stronger the interpretation ability of the input variables of the models to the inversion target. RMSE can reflect the deviation between retrieval values and real values and the value range of RMSE is (0, +∞). Its value will increase when the dispersion of the predicted value of the model is high. MAE is the mean of the absolute value of the error between the observed value and the predicted value. The value range of

MAE is (0, +∞). The higher the value of MAE, the poorer the predictive performance of the model. Thus, a model with high $R^2$, low RMSE, and low MAE is regarded as a qualified model for inversion.

## 3. Results

### 3.1. In Situ Data Analysis

The pollution level (NBO, MBO, SBO) of every sampling point can be determined according to the measurement of four parameters (DO, AN, SD, ORP) and the classification standards of black-odor water bodies shown in Table 2. Meanwhile, the NCPI of every sampling point can be calculated based on the field observation data. Results of both manual interpretation and NCPI are presented in Figure 4. The right *y*-axis stands for the pollution level with the regulation that 1 equals NBO, 2 equals MBO, and 3 equals SBO. The left *y*-axis represents the value of NCPI. It can be seen from the scatter plot of pollution levels that the first 48 points belong to NBO while the last four points belong to MBO. By observing the line graph, the NCPI of the first 48 are all lower than 1, in which case, these points belong to NBO in the light of the assumption in 2.5.1. The index value of the remaining points is between (1, 2), so they are judged as MBO. Since the result of NCPI corresponds with the result of manual interpretation, NCPI is fully competent in accessing the pollution level of water.

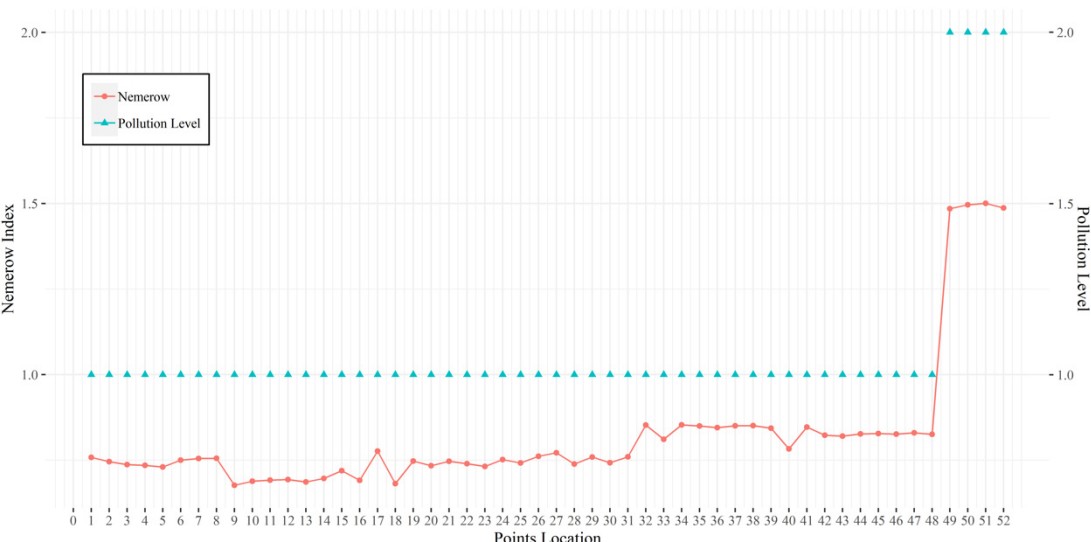

**Figure 4.** Manual interpretation of the pollution level and corresponding NCPI in study area.

### 3.2. Model Optimization and Accuracy Evaluation

Based on data processed in Section 2.4, XGBoost algorithm was chosen as one of the retrieval algorithms. Parameters, including gamma, eta, and the maximum number of iterations, were adjusted for the optimization of the model. The maximum number of iterations was accumulated in turn. The value of gamma was set to 0.001 and 0.1. The learning rate eta was determined appropriately. The inversion evaluation parameter$R^2$ of models with different parameters was shown in Figure 5. The iteration number was determined as 3250 based on Figure 5b. With the purpose of avoiding the over-fitting phenomenon and improving the model's generalization ability, the value of gamma cannot be too small while the learning rate eta should be reduced. After the adjustment above, the regression model with parameters (gamma = 0.001, eta = 0.3, the iteration number = 3250) was selected as the inversion model. On the training data, $R^2$ was 0.99, RMSE was 0.01, and MAE was 0.01. On the test data set, $R^2$ was lower than that of the train data set with a value of 0.94 and both RMSE and MAE increased a little, in which case, the XGBR model

l had remarkable generalization ability. It is worth noting that the difficulty of adjusting parameters is relatively low and the model is not easy to overfit.

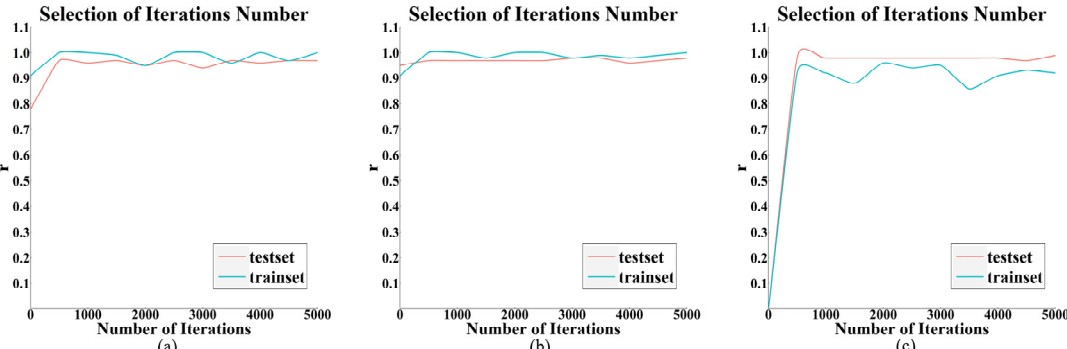

**Figure 5.** The change in $R^2$ based on XGBR model as the number iterations increases. (**a**) gamma = 0.001, eta = 0.6; (**b**) gamma = 0.001, eta = 0.3.; (**c**) gamma = 0.1, eta = 0.3.

Since the radial basis function was selected as the kernel function of SVR, parameters of significance were the penalty coefficient C and gamma. The higher the gamma value, the fewer support vectors and the easier the model is too overfit. The higher the C value, the less tolerance for errors and the easier the model is too overfit. Neither large C value nor small C value contributes to better generalization ability. The penalty coefficient C and gamma are determined to be 16 and 0.050 respectively. The $R^2$ of the training data set and the test data set reached 0.96 or more and, in which case, the performance of the SVR model was nearly the same as that of the XGBR model. However, compared with the XGBoost algorithm, it is difficult to generate an optimal model using the SVR due to the significant time cost and overfitting problem.

For RFR, the maximum feature number needs to be adjusted appropriately to avoid overfitting. Considering that the number of selected features and the sample size are small, the adjustment of hyperparameters, such as the maximum depth and the minimum sample number of the leaf nodes, has little impact on the model's accuracy. When the number of decision trees was set to 1000 and the maximum feature number was 11, the evaluation indicators tended to be stable without the occurrence of over-fitting or under-fitting. The model's retrieval accuracy on the test data sets and the training data sets is quite close and their $R^2$ both reached 0.87, which was lower than those of the other two models. The RMSE of the RFR model is the largest among the 3 models. Thus, the performance of the RFR model is inferior to that of XGBR.

The evaluation indicators of each model for NCPI are shown in Table 3. According to observations, the XGBR model had the best performance on the train data set with the highest $R^2$ and lowest error rate including RMSE and MAE. The prediction accuracy of its test data set was comparable to that of the train data set. For the train data set, the RMSE and MAE of the SVR model were higher than those of the XGBR model despite the fact that they have the same value of $R^2$. For the test data set, the prediction accuracy of the SVR model was slightly lower than that of the XGBR model. The retrieval accuracy of RFR model was significantly lower than that of the XGBR model since the RFR model had the lowest $R^2$ and highest RMSE on the train data set and test data set. Thus, both the XGBR model and SVR model can achieve higher inversion accuracy while the RFR model is the most inferior.

**Table 3.** Inversion accuracy of NCPI using different regression models.

| Modeling Method | Training Data | | | Test Data | | |
|---|---|---|---|---|---|---|
| | $R^2$ | RMSE | MAE | $R^2$ | RMSE | MAE |
| RFR | 0.87 | 0.09 | 0.05 | 0.87 | 0.10 | 0.05 |
| SVR | 0.99 | 0.02 | 0.02 | 0.92 | 0.09 | 0.09 |
| XGBR | 0.99 | 0.01 | 0.01 | 0.94 | 0.09 | 0.07 |

The scatter plot of estimated NCPI based on three regression models and observed NCPI is shown in Figure 6. The difference between the estimated NCPI and the observed NCPI indicates the model's prediction deviation. The significant difference implies that over-fitting or under-fitting phenomena occur. It can be seen that the estimated value of XGBR and SVR are more concentrated on the diagonal, in which case, predicted results deviate less from the true value. The difference between the predicted values and true values of the RFR model is large, which means that the prediction accuracy of the RFR model is poor.

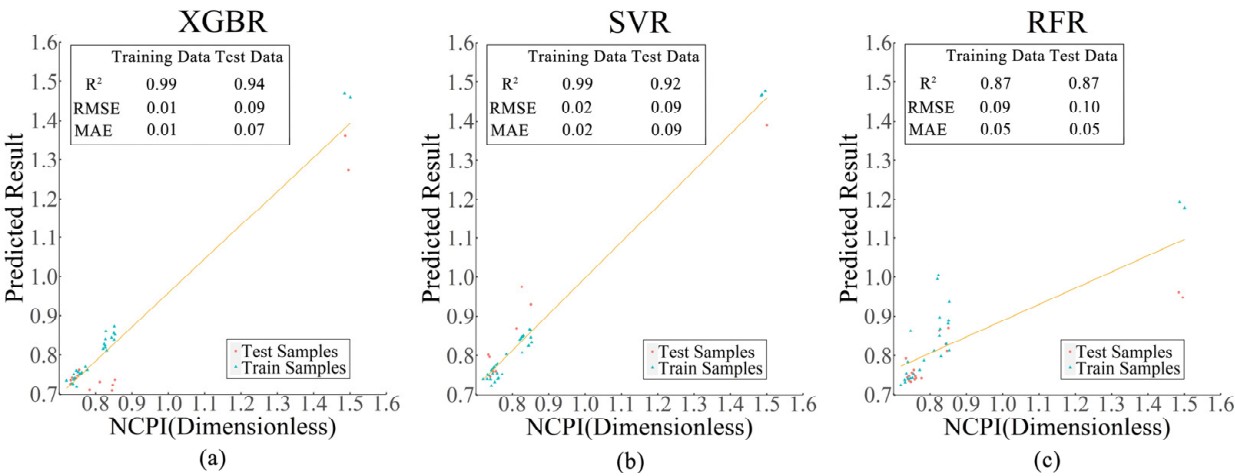

**Figure 6.** Scatter plot of the observed values and predicted values of NCPI using 3 regression models. (**a**) XGBR; (**b**) SVR; (**c**) RFR.

### 3.3. UAV-Borne Image Inversion Based on Three Models

Due to the comparison of accuracy in Section 3.2, it was found that all 3 regression models (XGR, SVR, RFR) with $R^2$ greater than 0.87 are qualified for inversion on UAV-borne multispectral images. The spectral average of a $5 \times 5$ pixel matrix was input into models to derive NCPI. Since true real values of NCPI range from 0.76 to 1.50, the inversion of effect will be considered poor if the predicted value is negative.

The inversion results based on three models were shown in Figure 7 providing a spatial depiction of NCPI. Since the values of NCPI from the area disturbed by the shadow and riverbed cannot reflect the real situation, they are not considered available for the statistical inversion results. The details are discussed in Section 4. Combined with the classification in Table 2, the black and odorous degree of water can be determined by the NCPI from Figure 7. According to the inverted map based on the XGBR model, the NCPI in 4 lakes is lower than 1, indicating that water bodies are NBO. For the lower reach of the Chebei river, the values of NCPI generally increase along the river, ranging from 0 to 1.503, which means water downstream of the Chebei river changes from non-black-odor to mild-black-odor.

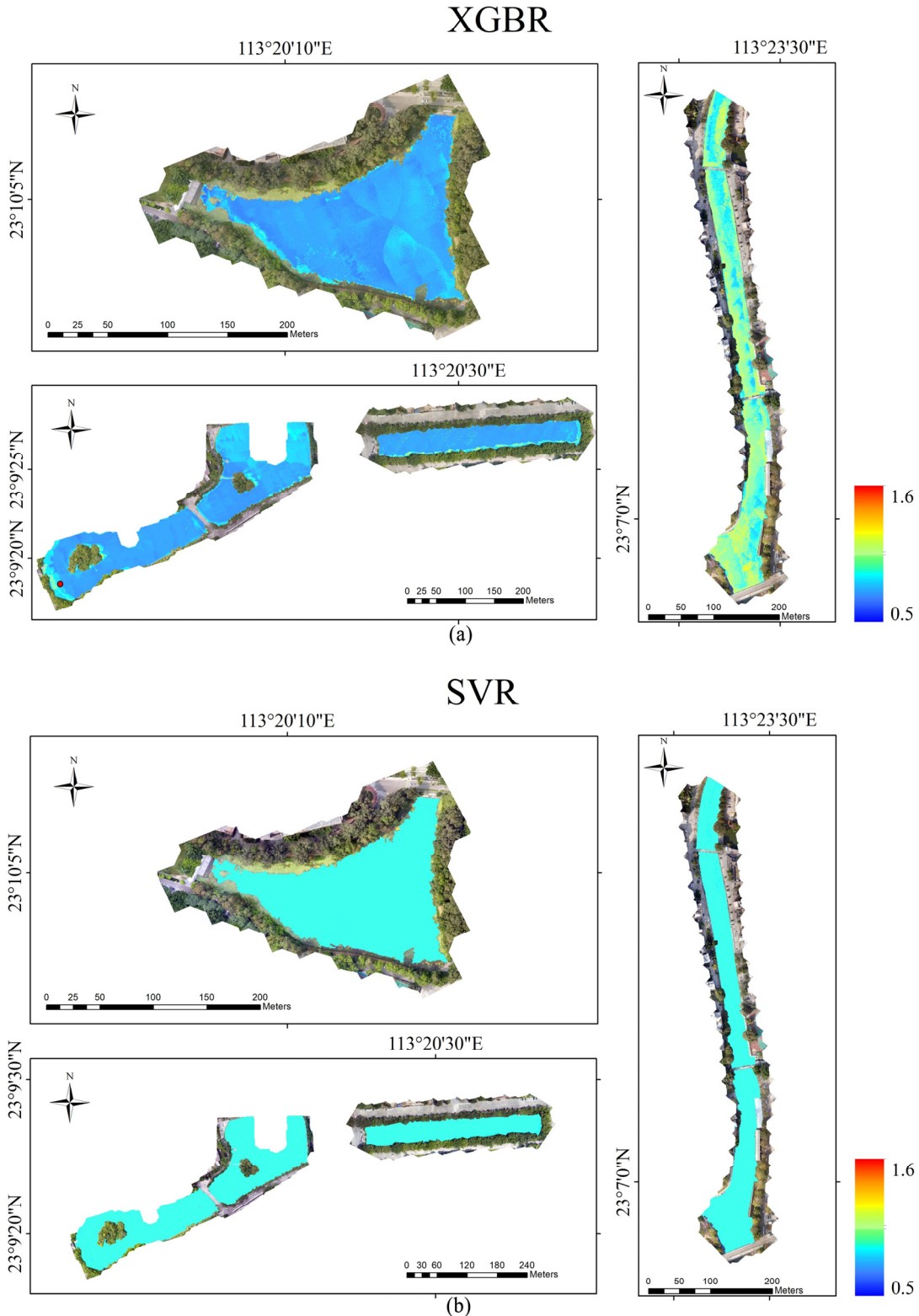

**Figure 7.** *Cont.*

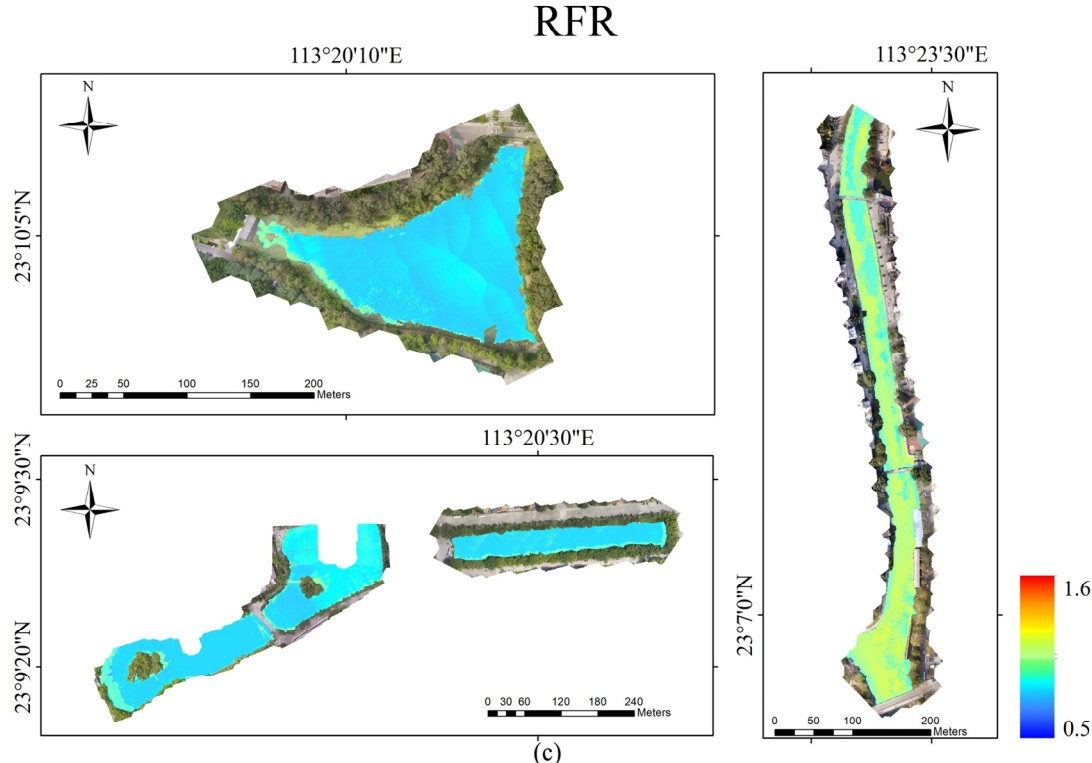

**Figure 7.** Inversion map of NCPI on the basis of UAV-borne images and 3 different models. (**a**) XGBR; (**b**) SVR; (**c**) RFR.

Statistical information of NCPI is shown in Table 4, where the first line is the real value (NCPI maximum = 1.50, NCPI minimum = 0.76) and the rest of the lines are various estimated values based on 3 models. The XGBR model performs best since the maximum and minimum values (maximum = 1.51, minimum = 0.62) of the estimated NCPI are both close to the real values. The retrieval result of the RFR model is slightly worse than that of XGBR since the predicted maximum value (maximum = 1.31) is lower than the real NCPI. The SVR model has the worst predictive effect with the narrow range of estimated values (maximum = 0.92, minimum = 0.82), in which case, the SVR model fails to distinguish between NBO and MBO.

**Table 4.** Statistical information of the UAV-borne image inversion based on the different models.

| Modeling Method | Computing Time (s) | Max Value | Min Value |
|---|---|---|---|
| In-situ Measurement | — | 1.50 | 0.76 |
| RFR | 130.7 | 1.31 | 0.74 |
| SVR | 109.4 | 0.92 | 0.82 |
| XGBR | 88.1 | 1.51 | 0.62 |

Additionally, calculating time varies from model to model, which is also shown in Table 4. Based on the same computer hardware and the same size of input data, the XGBR model requires the least fitting time at 88.1 s while the calculation time of the RFRR model is 1.47 times that of the XGBR model. The calculation speed of the RFR model is lower than that of the XGBR model but higher than that of the RFR model.

In summary, the XGBR model achieves the highest accuracy in the shortest time and the prediction results are the most consistent with field observation when it is used to invert NCPI based on the UAV-borne images. Although the RFR model obtained similar inversion results to the XGBR model, the operation time is much longer than the XGBR

model. This is an important concern when dealing with large volumes of image data. The SVR is not suitable for the inversion of NCPI since it is unable to invert high values.

## 4. Discussion

In the study, we established models for the retrieval of water parameters using UAV-borne multispectral images based on machine learning algorithms. According to the comparison in Section 3, the XGBR model is regarded as the most suitable model for the inversion of NCPI, due to its extremely high fitting accuracy on the training data set and test data set, the shortest operation time, and accurate inversion results on multispectral images. Our results are consistent with previous reports [53,69]. Lu et al. [53] found that the XGBR model had a more stable and satisfactory performance for predicting water quality parameters (Chl-a and suspended solid) compared with the SVR model and the RFR model. Research on water depth inversion of inland water bodies based on remote sensing and machine learning algorithms shows that the RFR model and the XGBR model had better predictive performance than the SVR model [69].

The Pearson correlation coefficient is used to measure the linear correlation between two variables. The closer the absolute value of the coefficient is to 1, the stronger the correlation between the 2 variables. In Section 2.4, it was found that the Pearson correlation coefficients between NCPI and band combinations are all less than 0.55. When the target variable is a single water quality parameter, the maximum Pearson coefficient between ORP and band combination is 0.48, which is slightly lower than that of other parameters (AN, SD, DO, ORP). The correlation analysis shows that neither NCPI nor a single water quality parameter has a strong linear correlation with the combination of bands. When inverting NCPI based on the MLR algorithm, stepwise regression method is applied to avoid the problem of multicollinearity and ensure that the values variance inflation factor values of the interpretation variable are less than 10 [70]. The retrieval accuracy of the training data set ($R^2$ = 0.36, RMSE = 0.10, MAE = 0.06) and test data set ($R^2$ = 0.40, RMSE = 0.25, MAE = 0.15) is much lower than that of machine learning algorithms (RFR, SVR, XGBR). The conclusion is the same as the result of Wei et al. [54]. For variables with weak linear correlation, XGBR is more applicable for establishing an inversion model of water quality since it enables to explore nonlinear relationships.

The reflectance data from the area near the lakeshore were disturbed by the presence of shadows due to trees along the shore. Moreover, the reflectance data from the Chebei river were disturbed by bare riverbeds and shadows of trees and buildings along the river. The shape of shadows and riverbeds is shown in Figure 8 based on the XGBR model, in which case, the disturbed area can be easily identified manually combined with remote sensing orthophoto maps. However, the sensitivity of the SVR model is not sufficient to show the interference in their inversion maps. In the study of Wen et al. [69], the predicted map of water depth based on XGBR reflects more features than that based on the SVR model. The reflectance data are unavoidably disturbed by various shadows or exposed water bottom. In this case, the ability of the XGBR model to identify the disturbed area is of significance, which makes it more qualified for water parameter inversion.

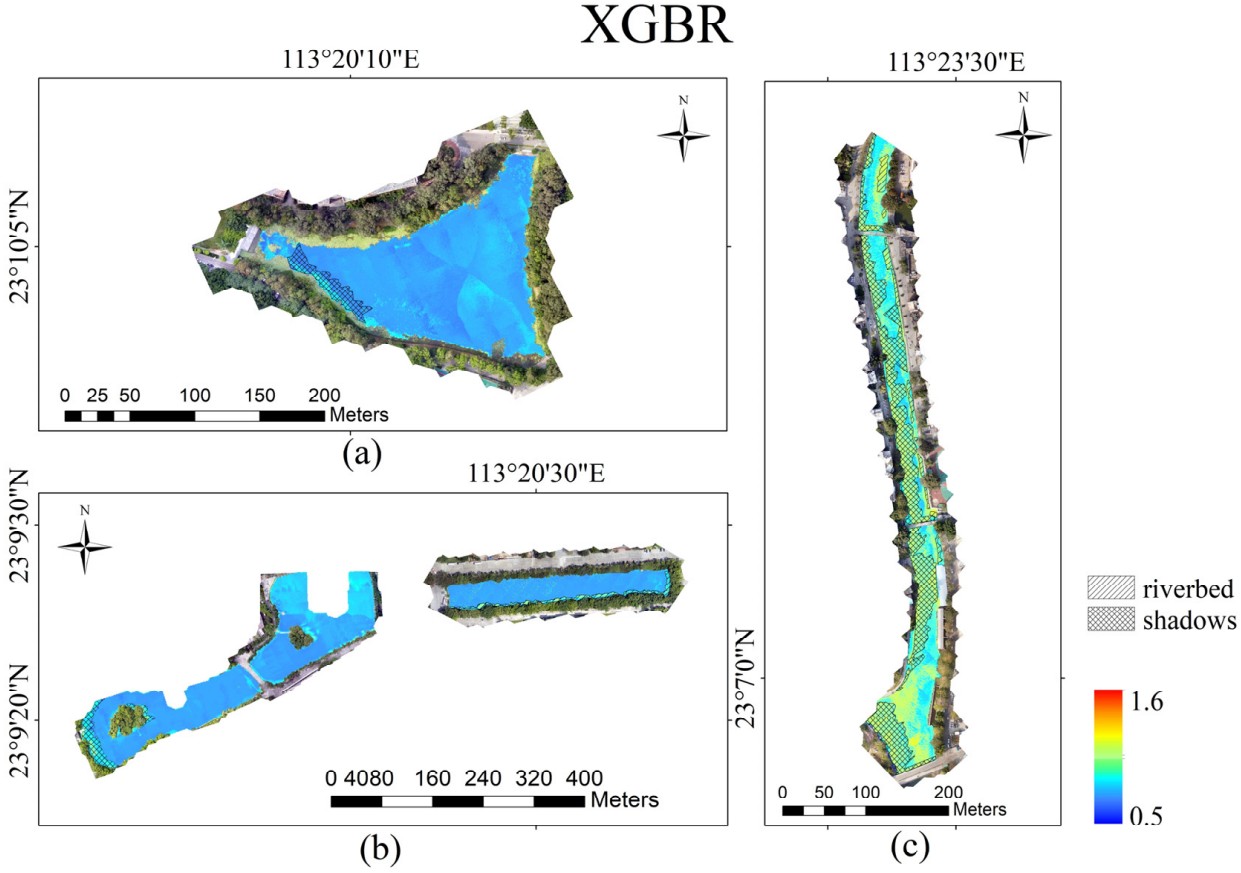

**Figure 8.** Shadow and rare riverbed in inversion map based on the XGBR model. (**a**) North Lake. (**b**) West Lake and Middle Lake (left); East Lake (right). (**c**) Chebei River.

The inversion map based on the XGBR model reveals the spatial distribution of NCPI in the study area. The NCPI of points 1–42 is blue, indicating that the water in four lakes is NBO, which shows concordance between simulations and observations. Except for the area disturbed by shadows and bare riverbeds, the NCPI of points 43–48 is light blue or green, indicating that these points in the Chebei river are NBO. Most of the remaining points are yellow, indicating that most of them are MBO. The value of NCPI is relatively high in the lower reaches of the Chebei river compared to that of four lakes which means the pollution level of the Chebei river is more severe than that of four lakes. In 2016, the Chebei river was identified as a MBO water body, which made it one of the 147 black and odorous water bodies listed in the national regulatory platform in Guangzhou. Various comprehensive measures were adopted to deal with the water quality problem, including engineering approaches (pollution source control, sewage interception, dredging, water supplement ecological restoration), and management approaches (the establishment of a system of "river chiefs" and so on). For example, the Three river Water Replenishment Project was proposed to solve the problem of water shortage in Chebei river during the dry season and after sewage interception. After treatment and remediation for years, most black-odor water bodies were eliminated in the Chebei river, and the water quality has been greatly improved. It can be seen from the inversion map based on the XGBR that most downstream area of the Chebei river is NBO, which is consistent with the real situation.

Since ORP, AN, and DO are non-optical parameters among the four parameters represented in Table 2, NCPI has weak optical characteristics. It is hard to correlate the spectral bands with non-optical water quality indicators since non-optical water quality indicators do not directly present optically diagnostic signals in water leaving radiance [71,72]. To obtain strong correlations between reflectance and NCPI, this study introduced band ratios

and band combinations commonly used in satellite remote sensing inversion. As a result, not only does this method improve the linear relationship between NCPI and variables, but some band ratios and band combinations are also of great significance to the modeling based on the RF and XGBoost algorithm. It can be observed that "B2/B3", "Kab", "B3/B1", "SABI", and "B1/B2" ranks in the top five among all variables in the RFR model according to % IncMSE (the prediction error of the model caused by randomly replacing the value of this variable). According to the evaluation indicator weight (the number of times a feature is used to split the data across all trees), the five most important variables for the XGBR model are "Kab", "SABI", "B1/B3", "B2/B3", and "B1/B2". The finding that band ratios contribute to generating inversion models capable of predicting non-optical parameters is in accordance with the study conclusion of Juan et al. [73].

During the process of reflectance data acquisition, environmental factors such as solar altitude angle matter. UAV should work when the solar elevation angle is greater than 30 degrees but not too large. The greater the sun elevation angle, the stronger the light intensity of the sun. Since the water surface is relatively smooth under intensive light conditions, the water body will manifest mirror-like reflections, in which case, the affected photos fail to record all details. After mosaicking, a blank area may appear in the whole remote-sensing image. Thus, data collection using UAV should possibly avoid midday operation from 11 a.m. to 1 p.m, which is also recommended by Su et al. [24]. The calculation of reflectance didn't take the skylight into account in the process of model establishment and inversion, which may have a negative impact on the accuracy of retrieval. One limitation of modeling was the uneven distribution of data utilized for the training model (the sample points of NBO outnumbered the sample points of MBO), which influences the robustness of the inversion model. Since there are no SBO sample points in the study area, the predictive result for SBO water bodies cannot be verified. We suggested that all types of data should be collected as much as possible to ensure the sample size and diversity of data. It is also necessary to ensure the data are distributed evenly. These methods are adopted to make the fit structurally robust and make the model more reliable.

## 5. Conclusions

This study evaluated the pollution degree of black-odor water based on UAV-borne multispectral images and artificial intelligence algorithms. Through this study, the following conclusions can be drawn:

(1) The XGBR model achieved a higher inversion accuracy than other traditional machine learning algorithms when modeling the nonlinear relationship between NCPI and band reflectance combinations. The values of $R^2$ on the training and test dataset both reached 0.94 or higher. The RMSE was 0.01 and 0.09, respectively and the values of MAE were both lower than 0.07. The SVR model was the second best-fitting model with high values of $R^2$ on the training dataset ($R^2 = 0.99$) and test dataset ($R^2 = 0.92$). The RFR model performed worst since it had the lowest $R^2$ on the training dataset ($R^2 = 0.87$) and test dataset ($R^2 = 0.87$).

(2) Among three regression models, the XGBR model is most suitable for the current scene due to the following two reasons. First, the XGBR model obtained the highest fitting accuracy. Second, the inversion results of XGBR were most consistent with the field observations and the calculation time is the shortest at 88.1 s. Neither the SVR model nor the RFR model could predict high values well since the inverted NCPI values based on them were smaller than the real values. Moreover, the calculation time of the RFR model is too long, which is not acceptable in engineering applications. Considering both inversion accuracy and efficiency, it is effective to employ the XGBR model for quantitative remote sensing of water assessment parameter NCPI.

(3) The NCPI spatial distribution map was generated according to the inversion results based on the UAV-borne multispectral data and the XGBR model. It can be observed from the map that most of the study area was not black and odorous. The area with higher pollution level mainly existed downstream of the Chebei river and it was MBO.

The inversion map can be used to monitor the changes in water quality in urban rivers for conducting water quality maintenance or treatment.

It can be seen that XGBoost showed great potential in the field of water quality retrieval. Thus, in future research, we will continue to explore the feasibility of other boosting algorithms, e.g., Catboost and Adaboost, in water quality inversion using UAV-borne spectral data. Additionally, we will try to combine multi-temporal inversion images to analyze the dynamic change of our study area for sustainable planning. Owing to the satisfactory inversion accuracy in our study, we highly recommend the combination of latest machine learning algorithms and UAV-borne data for monitoring urban black-odor water bodies.

**Author Contributions:** Conceptualization, X.L.; methodology, X.L. and F.W.; software, Y.L. and F.W.; validation, X.L.; formal analysis, Y.L. and F.W.; investigation, F.W.; resources, X.L., D.W., J.J. and H.H.; data curation, Y.L. and D.W.; writing—original draft preparation, F.W.; writing—review and editing, X.L. and J.J.; visualization, Y.L.; supervision, X.L. and H.H.; project administration, X.L.; funding acquisition, H.H. All authors have read and agreed to the published version of the manuscript.

**Funding:** The research is financially supported by the National Key R&D Program of China (2021YFC3001000), the National Natural Science Foundation of China (51879107, 52109019), the Science and Technology Planning Project of Guangdong Province in China (2020A0505100009).

**Data Availability Statement:** Not applicable.

**Conflicts of Interest:** The authors declare no conflict of interest.

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
