# Peer review of "Monitoring of Urban Black-Odor Water Using UAV Multispectral Data Based on Extreme Gradient Boosting"

_water, doi:10.3390/w14213354_

Round 1

Reviewer 1 Report (Previous Reviewer 2)

Accepted

Author Response

Please refer to the attachment "response letter_R1.doc" for details of the modification.

Reviewer 2 Report (Previous Reviewer 1)

In general, the authors answered all my questions and the quality of the manuscript has improved considerably. Some minor issues could be further improved. It is recommended that some figures be labeled a, b, c, etc., for clarity, such as  Figure 1, Figure 7 and Figure 8. Higher quality figures are needed, such as Figure 5.

Author Response

Please refer to the attachment "response letter_R2.doc" for details of the modification.

Reviewer 3 Report (New Reviewer)

Comments on the ms. Submitted to Water by Fangyi et al. entitled: “Monitoring of urban black-odor water using UAV multispectral data based on extreme gradient booting

General comments

This paper makes a nice contribution to the environmental study of urban rivers based on the construction of models relating spectral signatures of waters obtained by a multispectral sensor onboard a UAV and physical-chemical parameters directly measured in water. Although the idea is not novel at all, the usefulness of the application to black-odor waters in urban areas and the possibility of mapping the areas of the water bodies affected by this problem by simply flying a multispectral camera on top, deserves the effort.

The paper is well organized and in a couple of rounds is understandable. But could significantly benefit from both: summarizing the information in certain sections such as 2.2 (page 7, parts 2, 3, 4 and 5); removing some unnecessary tables such as table 1; improving the quality of certain figures such as fig. 2, 3, 5 and 6; and finally, reviewing the English stile throughout the ms.

Regarding the core of the paper, there are several issues the authors should attend and discuss:

(1) The election of the physical-chemical parameters to be measured in order to build the models. Authors choose Sechi disk depth (SC), ORP, ammonia nitrogen (AN) and dissolved oxygen (DO), but surprisingly no other parameters such as H2S concentration and electrical conductivity (EC) that do tell a lot about the health of the waters, especially in relation to black-odor. In relation to this, something that is missing in the ms. and that is essential, is to offer a geochemical characterization of the waters. It would help quite a lot to know the complete characterization of this waters. For example, what is the biochemical oxygen demand (BOD) which will tell us about the amount of O consumed by bacteria and microorganisms under aerobic conditions; or the chemical oxygen demand (COD) which will inform us on the oxygen necessary to oxidice the inorganic nutrients and organic material. Furthermore, what is the geological context of the river catchment, what are the parameters affecting its geochemical composition? All this info is relevant to your study and should be supplied.

(2) To me, a key issue is to know the data in such a way that at least some elemental calculations could be checked. I do not mean I want to reproduce your models, but, although I am sure you calculated well the Nemerow index, I would like to review this. Data are totally missing in your ms., both the physical-chemical parameters and the spectral results from the multispectral sensor. Authors should be make them available in a convenient way (as appendix, in electronic format, whatever the journal dictates). It is nice to cook, but it is totally necessary to know the ingredients of your dish!

(3) There is something in the discussion section that is not clear. Why the inversion accuracy of the training data set and test data is similar (if not better) in SVR respect XGBR but later on, the inversion map show a much worse result for SVR? You should explain this.

(4) Regarding the shadow along shore (page 16, lines 537 and so on), this discussion is irrelevant to the study. Just make a mask and remove that parts.

(5) Finally, you mention at the end of the conclusions that “Owing to the stable and satisfactory performance of XGBR in this study, we highly suggest that this approach could be employed to monitor black-odor water in other areas”. Do you mean that your specific models could be exported to other locations?  Or just that the general idea of making such a models considering the specific characteristics of every river could be used? Please make this clear, because it is not generally the case.

Additional comments

Page 2, line 46. “The Guide” needs a reference

Page 2, line 55. AS for…not As of…

Page 4, line 112, define all the acronyms, RFR in this case and so the rest.

Page 4, line 118. To which selecting parameters are you referring?

Page 4, lines 130-134. You should offer a more detail explanation on bases of the XGBoost algorithm?

Page 5, figure 2. In the Chebei river map, please, place the station numbers in red outside the map to make them visible.

Page 6, table 1. I think you can easily put this information in a line of the text and not to occupy space with this table that is not necessary.

Page 6, line 236. Why not to use the term “mosaicking” instead of inlaying which sounds rare.

Page 7, lines 237 to 263. This should be summarized without loosing any important information.

Page 7, line 278. What do you mean by “the exhaustive method”? Could you explain that?

Page 9, figure 3. It is not understandable what do you want to show in this figure. How do you calculate the dimensionless functions for SD, Do, ORP and AN?

Page 9, lines 313-314 What are the weight factor syou have use to calculate the Nemerow index? On the basis of what have you choosen the concrete Wi? This is a major point you should discuss.

Page 10, line 338. If the choose of the hyperparameters is important, you have to discuss what they are and what is their meaning.

Page 12, figure 5. Make the fonts bigger as figures are not visible.

Page 12, line 424-426. It is really calculation time an issue here?. In your table the time difference in calculation time between SVR and XGBR is about 80 seconds? Is this relevant or I am missing something?

Page 13, lines 455-457 and figure 6. Please review this information. You mention that the difference between predicted values and true values of the RFR model is large, but in figure 6, values are exactly the same. This is probably and error, or not? Please check this and discuss if necessary.

Page 13, line 473. You mean that the four lakes are NBO, not MBO as stated.

Page 15, line 485- The first sentence is repeated. Please delete.

Page 17, line 564. Typo error. Correct.

Author Response

修改内容详见附件“response letter_R3.doc”。

Round 2

Reviewer 3 Report (New Reviewer)

Authors have made and effort and discuss and correct to issues raised during the review, so I feel the ms. could be ready for publishing.

This manuscript is a resubmission of an earlier submission. The following is a list of the peer review reports and author responses from that submission.

Round 1

Reviewer 1 Report

This manuscript illustrates the use of UAV multispectral data and XGBoost algorithm to assess urban black-odor water. This study has a certain application in the field of urban water quality inversion with high resolution and accuracy. Essentially, this study inverted black-odor water based on the empirical relationship between spectral reflectance and in-situ measurements, which implied that the results of the study would vary depending on the actual measurements, such as the number of ground samples. In fact, the extraction and analysis of the feature spectral information are more meaningful for black-odor water inversion. Unfortunately, this manuscript excessively focuses on the description and analysis of the algorithm and lacks an in-depth analysis on the mechanism of water quality inversion.

Major comments:

1. Introduction.

L116-128: The description of multiple nonlinear algorithms focused more on model performance. However, the models mentioned in this study are basically black box models, which are very limited in explaining the mechanism of water quality inversion except for accuracy. It is recommended to add the study progress reflecting the spectral information analysis of water quality characteristics.

2. Methodology.

Section 2.6.2: It is highly recommended to simplify the description related to the XGBoost algorithm, most of which is routine knowledge.

3. Results.

Section 3.1: Where is Figure 4a?

L396-398: Where is the valid data support for this conclusion? There should be a comparison chart or table for the results.

Section 3.2. Many of the results are part of the methodological content. It is recommended to restructure and streamline the content of the results.

4. Discussion

In the discussion section, it is recommended to add the analysis about the contribution of spectral features to water quality inversion.

L525-528: For multiple linear regression models, the problem of multicollinearity needs to be considered.

Minor observation.

1. Figure 2 is not standardized, and it is suggested to label the serial number (a) (b) (c), etc., and make a description.

2. L203: The description of geometric correction and inlaying has not been mentioned in the next methods.

3. Table 1: Add citations and, in addition, add correlation coefficients if necessary.

4. Some expressions are not strict, such as R2.

5. The quality of the figures is very poor, and many numbers are not clear, such as Figure 6.

6. There are many problems with the language expressions, such as "so they requires..." (L12).

Reviewer 2 Report

This study evaluated the pollution degree of black-odor water based on UAV-borne 582 multispectral images and artificial intelligence algorithms. The structure of the manuscript is proper, the interesting discussion, the rich graphical support that makes the manuscript clearer, and the appropriately selected literature on the subject deserve praise. I have a few minor comments further to improve the quality and clarity of the manuscript as follows:

1. Introduction section is good but it should include similar cases and similar results achieved in other studies and areas.

2. Figure 2. Sampling points in study area, not clear.

3. In Figure 4, please add the North direction on the maps as well as the latitudes and longitudes lines.

4. The graphical support in this manuscript not clear.

5. In conclusion section, please add the  the possibilities for future research and study in this locations.